# Mendelian randomization study of whole blood viscosity and cardiovascular diseases

**Youngjune Bhak[1]\*, Albert Tenesa[1,2]\***

1 MRC Human Genetics Unit at the MRC Institute of Genetics and Molecular Medicine, University of Edinburgh, Western General Hospital, Edinburgh, United Kingdom, 2 The Roslin Institute, University of Edinburgh, Easter Bush Campus, Midlothian, United Kingdom

\* ybhak@ed.ac.uk (YB); Albert.Tenesa@ed.ac.uk (AT)

## Abstract

### Aims

Association between whole blood viscosity (WBV) and an increased risk of cardiovascular disease (CVD) has been reported. However, the causal relationship between WBV and CVD remains not thoroughly investigated. The aim of this study was to investigate the causal relation between WBV and CVD.

### Methods

Two-sample Mendelian randomization (MR) was employed, with inverse variance weighting (IVW) as the primary method, to investigate the casual relationship between WBV and CVD. The calculated WBV and medical records of 378,210 individuals participating in the UK Biobank study were divided into halves and analyzed.

### Results

The means of calculated WBVs were 16.9 (standard deviation: 0.8) and 55.1 (standard deviation: 17.2) for high shear rate (HSR) and low shear rate (LSR), respectively. 37,859 (10.0%) major cardiovascular events (MACE) consisted of 23,894 (6.3%) cases of myocardial infarction (MI), 9,245 (2.4%) cases of ischemic stroke, 10,377 (2.7%) cases of revascularization, and 5,703 (1.5%) cases of coronary heart disease-related death. In the MR analysis, no evidence was found indicating a causal effect of WBV on MACE (IVW p-value for HSR = 0.81, IVW p-value for LSR = 0.47), MI (0.92, 0.83), ischemic stroke (0.52, 0.74), revascularization (0.71, 0.54), and coronary heart disease-related death (0.83, 0.70). The lack of sufficient evidence for causality persisted in other MR methods, including weighted median and MR-egger.

### Conclusions

The Mendelian randomization analysis conducted in this study does not support a causal relationship between calculated WBV and CVD.

**Data Availability Statement:** Our data is available as a part of the UK Biobank project. Details of procedures for accessing the UK Biobank data can be found here: https://www.ukbiobank.ac.uk/enable-your-research/apply-for-access.

**Funding:** This project was funded by the National Institute for Health Research (NIHR) Artificial Intelligence and Multimorbidity: Clustering in Individuals, Space and Clinical Context (AIM-CISC) grant NIHR202639. The views expressed are those of the author(s) and not necessarily those of the NIHR or the Department of Health and Social Care. The funders had no role in study design, data collection and analysis, decision to publish, or preparation of the manuscript.

**Competing interests:** The authors have declared that no competing interests exist.

## Introduction

Cardiovascular diseases (CVD), including heart attack and stroke, are one of leading causes of morbidity and mortality globally [1]. Association between the risk of CVD and Whole blood viscosity (WBV), a measure of the thickness and flow resistance of bulk blood, have been reported [2–8]. However, establishing a causal relationship between WBV and CVD remains challenging due to the potential biases from confounding factors in traditional studies lacking randomized trial designs.

Mendelian randomization (MR), an epidemiological method, utilizes genetic variants robustly associated with an exposure of interest as instrumental variables (IVs) to investigate the causal effects of risk factors on specific outcomes [9]. The advantage of MR lies in the random assignment of these genetic variants at conception, rendering MR studies less susceptible to confounding factors compared to traditional observational studies [10]. Furthermore, MR is robust against reverse causality since the development of diseases does not alter individuals' genotypes. MR have been utilized to investigate casual relationships of risk factors such as blood pressure [11,12], obesity [13], type 2 diabetes mellitus [14] and, profile of blood lipids [15–17] in and CVD. The objective of this study was to use MR to examine the causal relationship between WBV and CVD. WBV of individuals was calculated by applying the formula previously reported [18].

## Methods

### Participants

The UK Biobank (UKB) is a prospective research resource of population-based cohort study that include comprehensive phenotype and genotype data from approximately 500,000 participants recruited in 2006–2010 residing in England, Scotland, and Wales (www.ukbiobank.ac.uk). This an open-access resource was established to support investigations into the factors influencing various health outcomes [19].

### Ethics statement

The UK Biobank project was approved by the National Research Ethics Service Committee North West-Haydock (REC reference: 11/NW/0382). Participants provided written informed consent to participate in the UK Biobank. An electronic signed consent was obtained from the participants. This research was conducted using the UK Biobank Resource under project 44986.

### Extrapolation of whole blood viscosity

WBV was calculated for both low shear rate (LSR) (0.5 sec$^{-1}$) and high shear rate (HSR) (208 sec$^{-1}$) from hematocrit (HCT) and total plasma protein concentration (TP) using the validated formulation [18]. HCT was calculated by multiplying red blood cell count by the mean corpuscular volume.

$$\text{HSR} : \text{WBV} \left(208\,\text{sec}^{-1}\right) = (0.12 \times \text{HCT}) + (0.17\,\text{TP}) - 2.07$$

$$\text{LSR} : \text{WBV} \left(0.5\,\text{sec}^{-1}\right) = (1.89 \times \text{HCT}) + (3.76\,\text{TP}) - 78.42$$

### Study outcomes

The data pertaining to each component of the participants' outcomes in the present study was accessible through the UK Biobank study [19]. The primary outcome of the study was major

cardiovascular events (MACE), which encompassed a composite outcome involving the occurrence of non-fatal MI, coronary revascularization (defined as "percutaneous transluminal coronary angioplasty, PTCA" or "coronary artery bypass grafting, CABG"), ischemic stroke, or death due to coronary heart disease (CHD).

These specific outcomes were defined and categorized as follow: non-fatal MI defined algorithmically by UK Biobank (ICD9: 410.X, 411.0.X, 412.X, 429.79; ICD10: I21.X, I22.X, I23.X, I24.1, I25.2; self-report 20002: 1075), PTCA or CABG (self-report 20004: 1070, 1095, 1523; Procedures (OPCS): K50.1, K40.X, K41.X, K42.X, K43.X, K44.X), ischemic stroke (ICD9: 434. X, 436.X; ICD10: I63.X, I64.X; self-report 20002: 1583), and death due to CHD (Death 40001, 40002: I21.X, I22.X, I23.X, I24.X, I25.1, I25.2, I25.3, I25.5, I25.6, I25.8, I25.9) [17].

## Mendelian randomization

We conducted split sample approach in two-sample MR setting to avoid sample overlap [20].

The dataset was randomly split into two halves, and GWASs were performed to estimate both instrumental variable–exposure and instrument variable–outcome associations for each half.

To ensure homogeneity, we limited our analyses to unrelated individuals of White British ancestry. Additionally, individuals with more than 10% missing genotypes or those with discrepancies between recorded sex and genetically determined sex were excluded. Following these exclusions, the final dataset consisted of 378,210 participants (Table 1).

GWASs were conducted using genotypes of the unrelated White British individuals. White British individuals are inferred from UK Biobank records. The unrelated individuals were identified using the KING software with following options: - -unrelated - -degree 2 (version 2.2.8) [21]. Autosomal genotypes of unrelated White British were further filtered using PLINK software (version 1.90p) with the following options; - -geno 0.01, - -hwe 1e-15, - -maf 0.01, and - -mind 0.1 [22]. For the GWASs of WBVs, REGENIE software (version 3.2.2) was utilized with the following option; - -apply-rint [23]. Covariates considered in the GWAS included age, age square, genetic principal components 1 to 20, sex, and genotyping array.

To select independent instrumental variants for Mendelian randomization, summary statistics were clumped to extract index variants using PLINK software with the following options; - -clump, - -clump-p1 0.00000005, - -clump-r2 0.001, and - -clump-kb 10000 [22]. To avoid the risk of weak instrument bias, variants with F-statistics $> 10$ were selected [24]. The variants were filtered out if a variant had a reported association with CVD and or factors related with blood viscosity or CVD. The associations were investigated by utilizing PhenoScanner database with the following options; catalogue: diseases & traits, p-value: $5 \times 10^{-8}$, proxies—EUR, $r^2$: 0.8, and build: 37 [25,26]. Such factors include obesity [13,27–29], blood pressure [11,12,30], lipid traits [15–17,31–35], type 2 diabetes mellitus [14,36,37], smoking and alcohol intake [38–40]. For HSR, 45 variants from one half split and 37 variants from the other half split passed the filters, respectively. For LSR, 49 variants from one half split and 41 variants from the other half split passed the filters, respectively (S1–S4 Tables).

We conducted MR analysis to derive causal estimates using the TwoSampleMR R package (version 0.5.7) [41]. The causal estimates were initially derived using the inverse-variance weighted (IVW) method [42], followed by weighted median (WM) [43] and, MR-egger methods [44]. We estimated intercept of MR-Egger to test horizontal pleiotropy [45] and Q statistics to test global heterogeneity of the genetic instruments [46,47]. The resulting estimates from each half were then combined with fixed-effect meta-analysis to give a single estimate.

**Table 1. Baseline characteristics.**

| Baseline Characteristics of the Participants.* | |
| --- | --- |
| Variable | Participants (N = 378,210) |
| Age (yr) | 57.0 ± 7.9 |
| Female sex (%) | 53.7 |
| Blood pressure (mm Hg) | |
| Systolic | 140.3 ± 19.7 |
| Diastolic | 82.3 ± 10.7 |
| Body-mass index | 27.4 ± 4.7 |
| Diabetes (%) | 4.8 |
| Current smoker (%) | 10.1 |
| Lipid levels (mg/dl) | |
| Total cholesterol | 221.0 ± 44.3 |
| LDL cholesterol | 138.1 ± 33.7 |
| HDL cholesterol | 56.2 ± 14.8 |
| Triglyceride level | 155.8 ± 90.6 |
| Whole blood viscosity | |
| LSR | 55.1 ± 17.2 |
| HSR | 16.9 ± 0.8 |
| Outcomes (%) | |
| Major cardiovascular event | 37,857 (10.0) |
| Myocardial infarction | 23,893 (6.3) |
| Ischemic stroke | 9,245 (2.4) |
| Revascularization | 10,376 (2.7) |
| Death due to CHD | 5,703 (1.5) |

Plus–minus values are means ±SD. HDL denotes high-density lipoprotein, LDL low-density lipoprotein, LSR low-shear rate, HSR high shear rate, and CHD coronary heart disease. Major cardiovascular event includes myocardial infarction, ischemic stroke, revascularization, and death due to CHD.

## Statistical analyses

Categorical variables were presented as counts and percentages, continuous variables were presented as mean and standard deviations (SD). All significance tests were two-tailed, and statistical significance was determined at $p < 0.05$. The statistical analyses were performed using the R version 4.2.1 (R Foundation for Statistical Computing, Vienna, Austria).

## Results

A total of 378,210 individuals were included in the study. The mean age ± SD was 57.0 ± 7.9 years. 53.7% of participants were women. The mean ± SD of calculated WBV were 55.1 ± 17.2 and 16.9 ± 0.8 for LSR and HSR, respectively. 37,859 (10.0%) of major cardiovascular events, including 23,893 (6.3%) myocardial infarction, 9,245 (2.4%) ischemic stroke, 10,376 (2.7%) revascularization, and 5,703 (1.5%) death due to CHD were recorded.

None of MR analyses provided evidence for a causal effect of WBVs on the risk of CVDs (Fig 1 and S5 Table). HSR didn't show causality for MACE (IVW estimates: -0.02, 95% Confidence interval: -0.16–0.13, P = 0.81), MI (0.01, -0.15–0.17, P = 0.92), stroke (0.07, -0.15–0.30, p = 0.52), revascularization (0.04, -0.18–0.26, P = 0.71), and death by CVD (0.03, -0.22–0.27, P = 0.83). The not significant causal estimate remained consistent with WM and MR-egger

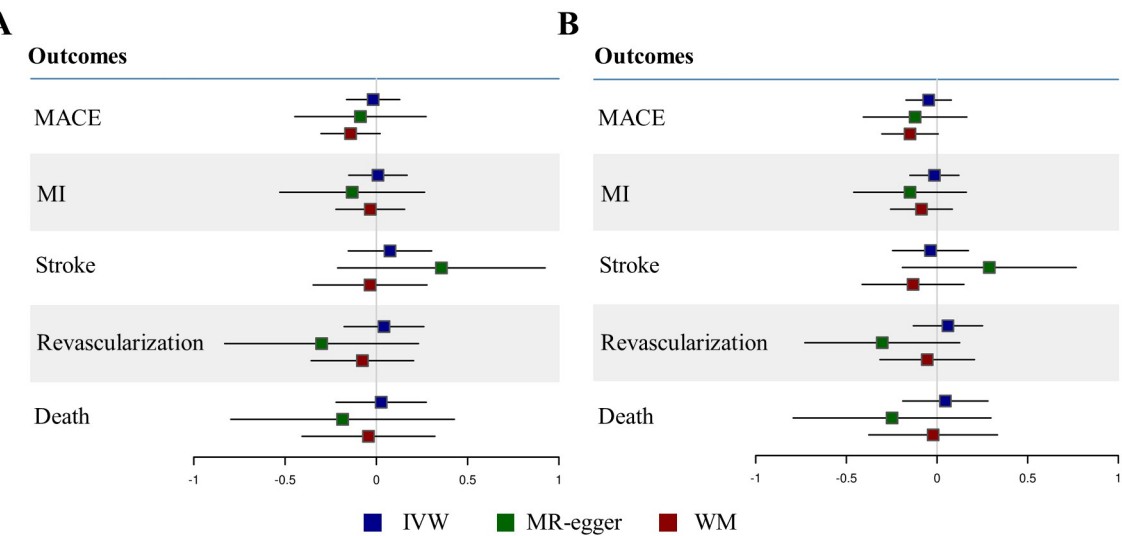

**Fig 1. Mendelian randomisation analysis between WBV and the risk of cardiovascular events.** Panel A shows mendelian randomisation results for the causal effect of HSR to the risk of cardiovascular events, and panel B shows mendelian randomisation results for the causal effect of LSR to the risk of cardiovascular events. The boxes represent causal estimates and the lines represent 95% confidence intervals. MACE denotes major cardiovascular event, MI myocardial infarction, IVW inverse-variance weighted, WM weighted median.

results (Fig 1A). LSR also didn't show causality for MACE (-0.05, -0.17–0.08, P = 0.47), MI (-0.01, -0.15–0.12, P = 0.83), stroke (-0.04, -0.25–0.17, P = 0.74), revascularization (0.06, -0.13–0.25, P = 0.54), and death by CVD (0.05, -0.19–0.28, P = 0.70). The not significant causal estimate remained consistent with WM and MR-egger results (Fig 1B).

## Discussion

This study used MR to investigate the potential causal association between WBV and CVD. The result from this study indicated insufficient evidence to substantiate a causal link between WBV and the risk of CVD.

WBV is susceptible to various influencing factors. Notably, WBV demonstrate non-Newtonian fluid behaviour. Under conditions of low shear rates, blood cells tend to aggregate, leading to an elevation in viscosity. Conversely at HSR, the opposite phenomenon occurs [48–50]. WBV has been noted to exhibit association with CVD and cardiovascular risk factors in previous studies [3,51,52]. However, upon adjustments for these risk factors, the association between WBV and CVD was found to be statistically non-significant in the subsequent study [53]. The not significant association following adjustments remained consistently evident in this study using MR.

However, the study result should be interpreted and considered carefully since we have not utilized directly measured WBV. While the formula employed in our study has undergone validation and has been applied in previous researches [18,54,55], it does not considered factors for WBV such as blood cell aggregability and deformability [56]. To establish a robust causal link between WBV and both CVD and CVD-related factors, future studies should incorporate measured WBV values, taking into account these critical variables.

## Supporting information

**S1 Table. Instrumental variants for HSR in Group 1.**
(XLSX)

**S2 Table. Instrumental variants for HSR in Group 2.**
(XLSX)

**S3 Table. Instrumental variants for LSR in Group 1.**
(XLSX)

**S4 Table. Instrumental variants for LSR in Group 2.**
(XLSX)

**S5 Table. Mendelian randomization estimates.**
(XLSX)

## Acknowledgments

This work used the Edinburgh Compute and Data Facility (ECDF) (http://www.ecdf.ed.ac.uk/). This research has been conducted using the UK Biobank Resource project 44986. For the purpose of open access, the author has applied a CC-BY public copyright licence to any Author Accepted Manuscript version arising from this submission.

## Author Contributions

**Conceptualization:** Youngjune Bhak.

**Data curation:** Youngjune Bhak.

**Formal analysis:** Youngjune Bhak.

**Funding acquisition:** Albert Tenesa.

**Investigation:** Youngjune Bhak.

**Methodology:** Youngjune Bhak.

**Project administration:** Youngjune Bhak.

**Resources:** Youngjune Bhak.

**Software:** Youngjune Bhak.

**Supervision:** Albert Tenesa.

**Visualization:** Youngjune Bhak.

**Writing – original draft:** Youngjune Bhak.

**Writing – review & editing:** Youngjune Bhak, Albert Tenesa.

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
