## [Decision Letter · Decision Letter 0]

28 Nov 2023

PONE-D-23-33347

Mendelian Randomization Study of Whole Blood Viscosity and Cardiovascular Diseases

PLOS ONE

Dear Dr. Bhak,

Thank you for submitting your manuscript to PLOS ONE. After careful consideration, we feel that it has merit but does not fully meet PLOS ONE’s publication criteria as it currently stands. Therefore, we invite you to submit a revised version of the manuscript that addresses the points raised during the review process.

We look forward to receiving your revised manuscript.

Kind regards,

Eyüp Serhat Çalık

Academic Editor

PLOS ONE

Additional Editor Comments:

Dear Authors

Your article on the relationship between whole blood viscosity and CVD offers new perspectives for researchers on the subject and highlights important points to be considered. Your manuscript has been evaluated by two distinguished reviewers and their recommendations are given below. I think that you can expand your manuscript, especially your discussion section, by using their suggested references. We are also looking forward to your manuscript after major revision by taking into account the second reviewer's suggestions on statistical analysis. I wish you success.

Reviewers' comments:

Reviewer's Responses to Questions

**Comments to the Author**

1. Is the manuscript technically sound, and do the data support the conclusions?

Reviewer #1: Yes

Reviewer #2: No

2. Has the statistical analysis been performed appropriately and rigorously? 

Reviewer #1: Yes

Reviewer #2: No

3. Have the authors made all data underlying the findings in their manuscript fully available?

Reviewer #1: Yes

Reviewer #2: Yes

4. Is the manuscript presented in an intelligible fashion and written in standard English?

Reviewer #1: Yes

Reviewer #2: Yes

5. Review Comments to the Author

Reviewer #1: The association between whole blood viscosity (WBV) and increased risk of cardiovascular disease (CVD) has been reported. In this study, the causal relationship between WBV and CVD was investigated.

I have a few suggestions;

1-" I recommend you to review the articles titled "The relationship between blood viscosity and acute arterial occlusion" and "Relationship between whole blood viscosity and lower extremity peripheral artery disease severity".

Reviewer #2: This study investigated the associations between a PRS for whole blood viscosity and major cardiovascular event (MACE) in the UK Biobank through a split sample approach. The selection of genetic instrument from GWAS is rigorous, using an allele score (PRS in this case) to collectively represent the genetic instruments is also valid, but the statistical analysis - linear or logistic regression using PRS and outcome is not technically an instrumental variable analysis (and thus not a Mendelian randomization study).

Using individual level data, MR should be preformed using two-stage least squares regression (see the Burgess et al. 2016 paper below). Otherwise, the authors could use the split sample approach in two-sample MR setting: using subset1 and subset2 to estimate the IV-exposure and IV-outcome associations, and repeat the analysis in the reversed way, finally, meta-analyse the results. The current statistical analysis only estimates the ASSOCIATION between genetic liabilities towards high/low WBV and observed MACE outcomes.

Burgess, S., Dudbridge, F., and Thompson, S. G. (2016) Combining information on multiple instrumental variables in Mendelian randomization: comparison of allele score and summarized data methods. Statist. Med., 35: 1880–1906. doi: 10.1002/sim.6835.

6. PLOS authors have the option to publish the peer review history of their article (what does this mean?). If published, this will include your full peer review and any attached files.

Reviewer #1: No

Reviewer #2: No

---

## [Author Response · Author response to Decision Letter 0]

30 Dec 2023

We have re-checked the journal’s style requirements and updated the manuscript as follows: 

*In the author list part, we changed affiliation indication to number.

*In the affiliation part, we removed ZIP or postal codes.

*In the corresponding authorship part, we excluded physical addresses and left only email addresses.

*In the manuscript, we have changed the font size and headings to the PLOS ONE template.

*In the manuscript, we have changed the citation of figures as Fig x.

*In the manuscript, we moved figure captions and table directly after the paragraph in which they are first cited.

The data presented here is available in the UK Biobank, as stated in the manuscript. No changes are needed for the Data Availability statement.

We have included captions for Supporting information files at the end of the manuscript and updated in-text citations to match accordingly.

Reviewers' comments:

Reviewer's Responses to Questions

Comments to the Author

1. Is the manuscript technically sound, and do the data support the conclusions?

Reviewer #1: Yes

Reviewer #2: No

We have modified manuscript to accommodate reviewers’ suggestions. (see below point-by-point responses)

2. Has the statistical analysis been performed appropriately and rigorously?

Reviewer #1: Yes

Reviewer #2: No

We have presented responses to accommodate reviewers’ comments. (see below point-by-point responses)

3. Have the authors made all data underlying the findings in their manuscript fully available?

Reviewer #1: Yes

Reviewer #2: Yes

4. Is the manuscript presented in an intelligible fashion and written in standard English?

Reviewer #1: Yes

Reviewer #2: Yes

5. Review Comments to the Author

Reviewer #1: The association between whole blood viscosity (WBV) and increased risk of cardiovascular disease (CVD) has been reported. In this study, the causal relationship between WBV and CVD was investigated.

I have a few suggestions;

1-" I recommend you to review the articles titled "The relationship between blood viscosity and acute arterial occlusion" and "Relationship between whole blood viscosity and lower extremity peripheral artery disease severity".

Thank you for sharing the articles; they significantly contribute to the manuscript, providing a deeper understanding and insight for the readers. We have appropriately referenced these articles in the introduction section (line 5, page 3, reference numbers 6 and 7).

Reviewer #2: This study investigated the associations between a PRS for whole blood viscosity and major cardiovascular event (MACE) in the UK Biobank through a split sample approach. The selection of genetic instrument from GWAS is rigorous, using an allele score (PRS in this case) to collectively represent the genetic instruments is also valid, but the statistical analysis - linear or logistic regression using PRS and outcome is not technically an instrumental variable analysis (and thus not a Mendelian randomization study). 

Using individual level data, MR should be preformed using two-stage least squares regression (see the Burgess et al. 2016 paper below). Otherwise, the authors could use the split sample approach in two-sample MR setting: using subset1 and subset2 to estimate the IV-exposure and IV-outcome associations, and repeat the analysis in the reversed way, finally, meta-analyse the results. The current statistical analysis only estimates the ASSOCIATION between genetic liabilities towards high/low WBV and observed MACE outcomes.

Burgess, S., Dudbridge, F., and Thompson, S. G. (2016) Combining information on multiple instrumental variables in Mendelian randomization: comparison of allele score and summarized data methods. Statist. Med., 35: 1880–1906. doi: 10.1002/sim.6835.

Thank you for providing your valuable insight into the methodology presented in the manuscript and offering guidance on the effective analysis method. In response to the reviewer’s comments, we have implemented the split sample approach in two-sample MR setting. The method section and the result section in the manuscript have been revised accordingly.

We retitled the Genetic instrument subsection to the Mendelian randomization and introduced the concept of split sample approach used in the study at the beginning of the section (from line 16 to 19, page 5). 

We screened genetic variants by querying them one by one, as an error was found when we queried groups of variants together. Consequently, we discovered additional variants with evidence of pleiotropy that were overlooked in the previous analysis. These newly identified variants were subsequently filtered out in the current analysis (from line 18 to 20, page 6).

We derived causal estimates of whole blood viscosity on outcomes using the inverse-variance weighted method, weighted median, and MR-Egger method. This was performed twice, separately using IV-exposure associations from subset 1 and IV-outcome associations from subset 2, and vice versa. A single estimate was then derived by combing the results from these two analyses (from line 21, page 6 to line2, page 7).

The resulting MR analysis still indicates insufficient evidence for a causal relationship between whole blood viscosity and the outcome. We have made corresponding modifications to the result section (from line 8 to 14), figure 1, and Supplementary Table 5 to present the updated MR results. We have removed the contents derived from the results of the previous association-based study

6. PLOS authors have the option to publish the peer review history of their article (what does this mean?). If published, this will include your full peer review and any attached files.

Do you want your identity to be public for this peer review? For information about this choice, including consent withdrawal, please see our Privacy Policy.

Reviewer #1: No

Reviewer #2: No

---

## [Decision Letter · Decision Letter 1]

29 Jan 2024

PONE-D-23-33347R1Mendelian Randomization Study of Whole Blood Viscosity and Cardiovascular DiseasesPLOS ONE

Dear Dr. Bhak,

Thank you for submitting your manuscript to PLOS ONE. After careful consideration, we feel that it has merit but does not fully meet PLOS ONE’s publication criteria as it currently stands. Therefore, we invite you to submit a revised version of the manuscript that addresses the points raised during the review process.

We look forward to receiving your revised manuscript.

Kind regards,

Eyüp Serhat Çalık

Academic Editor

PLOS ONE

Journal Requirements:

Reviewers' comments:

Reviewer's Responses to Questions

**Comments to the Author**

1. If the authors have adequately addressed your comments raised in a previous round of review and you feel that this manuscript is now acceptable for publication, you may indicate that here to bypass the “Comments to the Author” section, enter your conflict of interest statement in the “Confidential to Editor” section, and submit your "Accept" recommendation.

Reviewer #3: All comments have been addressed

Reviewer #4: All comments have been addressed

2. Is the manuscript technically sound, and do the data support the conclusions?

Reviewer #3: Yes

Reviewer #4: (No Response)

3. Has the statistical analysis been performed appropriately and rigorously? 

Reviewer #3: Yes

Reviewer #4: (No Response)

4. Have the authors made all data underlying the findings in their manuscript fully available?

Reviewer #3: Yes

Reviewer #4: (No Response)

5. Is the manuscript presented in an intelligible fashion and written in standard English?

Reviewer #3: Yes

Reviewer #4: (No Response)

6. Review Comments to the Author

Reviewer #3: I have reviewed the manuscript entitled 'Mendelian Randomization Study of Whole Blood Viscosity and Cardiovascular Diseases'.

The manuscript mentions the role of WBV in cardiovascular diseases. The thrombus presencen has been linked to WBV in patients admitted to outpatient clinic. Please mention this link in the introduction section citing 'Association of whole blood viscosity with thrombus presence in patients undergoing transoesophageal echocardiography'.

Reviewer #4: The authors’ responses have answered all earlier comments. I don't have any further comments on the current version.

7. PLOS authors have the option to publish the peer review history of their article (what does this mean?). If published, this will include your full peer review and any attached files.

Reviewer #3: No

Reviewer #4: No

---

## [Author Response · Author response to Decision Letter 1]

1 Feb 2024

Thank you for revieing the manuscript. We have checked the list of references and edited the list of authors in the reference “Relationship between diabetes mellitus and blood viscosity as measured by the digital microcapillary® system.”, from “Andrea V, Timan I, editors” to “Andrea V and Timan I”. The authors have confirmed that the manuscript doesn’t include papers that have been retracted and are happy to revise it if the journal could provide the list of papers that the issue arose. 

Reviewers' comments:

Reviewer's Responses to Questions

Comments to the Author

1. If the authors have adequately addressed your comments raised in a previous round of review and you feel that this manuscript is now acceptable for publication, you may indicate that here to bypass the “Comments to the Author” section, enter your conflict of interest statement in the “Confidential to Editor” section, and submit your "Accept" recommendation.

Reviewer #3: All comments have been addressed

Reviewer #4: All comments have been addressed

2. Is the manuscript technically sound, and do the data support the conclusions?

Reviewer #3: Yes

Reviewer #4: (No Response)

3. Has the statistical analysis been performed appropriately and rigorously?

Reviewer #3: Yes

Reviewer #4: (No Response)

4. Have the authors made all data underlying the findings in their manuscript fully available?

Reviewer #3: Yes

Reviewer #4: (No Response)

5. Is the manuscript presented in an intelligible fashion and written in standard English?

Reviewer #3: Yes

Reviewer #4: (No Response)

6. Review Comments to the Author

Reviewer #3: I have reviewed the manuscript entitled 'Mendelian Randomization Study of Whole Blood Viscosity and Cardiovascular Diseases'.

The manuscript mentions the role of WBV in cardiovascular diseases. The thrombus presencen has been linked to WBV in patients admitted to outpatient clinic. Please mention this link in the introduction section citing 'Association of whole blood viscosity with thrombus presence in patients undergoing transoesophageal echocardiography'.

Thank you for reviewing the manuscript and sharing the article; it significantly contributes to the manuscript, providing a broader understanding and invaluable insight for the readers. We have appropriately referenced the article in the introduction section (line 5, page 3, reference number 8).

Reviewer #4: The authors’ responses have answered all earlier comments. I don't have any further comments on the current version.

Thank you for reviewing the manuscript and providing your invaluable guidance. Your comments have enhanced the manuscript, offering a deeper understanding and invaluable insights for the readers. We greatly appreciate it.

7. PLOS authors have the option to publish the peer review history of their article (what does this mean?). If published, this will include your full peer review and any attached files.

Do you want your identity to be public for this peer review? For information about this choice, including consent withdrawal, please see our Privacy Policy.

Reviewer #3: No

Reviewer #4: No

---

## [Editor Report · Decision Letter 2]

5 Feb 2024

Mendelian Randomization Study of Whole Blood Viscosity and Cardiovascular Diseases

PONE-D-23-33347R2

Dear Dr. Bhak,

We’re pleased to inform you that your manuscript has been judged scientifically suitable for publication and will be formally accepted for publication once it meets all outstanding technical requirements.

Kind regards,

Eyüp Serhat Çalık

Academic Editor

PLOS ONE
---

## [Editor Report · Acceptance letter]

20 Mar 2024

PONE-D-23-33347R2 

PLOS ONE

Dear Dr. Bhak, 

I'm pleased to inform you that your manuscript has been deemed suitable for publication in PLOS ONE. Congratulations! Your manuscript is now being handed over to our production team.

Kind regards, 

on behalf of

Dr. Eyüp Serhat Çalık 

Academic Editor

PLOS ONE